# HPE: Answering Complex Questions over Text by Hybrid Question Parsing and Execution

**Ye Liu[1], Semih Yavuz[1], Rui Meng[1], Dragomir Radev[2],**
**Caiming Xiong[1], Shafiq Joty[1], Yingbo Zhou[1]**
[1] Salesforce Research, [2] Yale University
yeliu@salesforce.com

## Abstract

The dominant paradigm of textual question answering with end-to-end neural models excels at answering simple questions but falls short on explainability and dealing with more complex questions. This stands in contrast to the broad adaptation of semantic parsing approaches over structured data sources (e.g., relational database), that convert questions to logical forms and execute them with query engines. Towards the goal of combining the strengths of neural and symbolic methods, we propose a framework of question parsing and execution for textual QA. It comprises two central pillars: (1) parsing a question of varying complexity into an intermediate representation, named H-expression, which symbolically represents how an answer to the question can be reached by hierarchically combining answers from the primitive simple questions; (2) to execute the resulting expression, we design a hybrid executor, which integrates deterministic rules to translate the symbolic operations with a drop-in neural reader to answer each simple question. The proposed framework can be viewed as a top-down question parsing followed by a bottom-up answer backtracking. H-expressions closely guide the execution process, offering higher precision besides better interpretability while still preserving the advantages of the neural readers for resolving primitive elements. Our extensive experiments on four different QA datasets show that the proposed framework outperforms existing approaches in supervised, few-shot, and zero-shot settings, while also effectively exposing the underlying reasoning process[1].

## 1 Introduction

End-to-end neural models that transductively learn to map questions to their answers have been the dominating paradigm for textual question answering (Raffel et al., 2020; Yasunaga et al., 2021) ow-

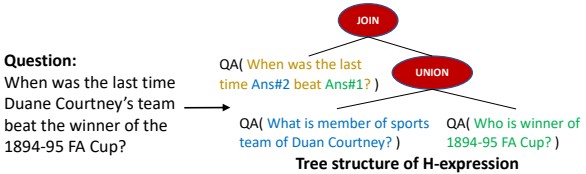

Figure 1: An illustration of H-expression.

ing to their flexibility and solid performance. However, they often suffer from a lack of interpretability and generalizability to more complex scenarios. Symbolic reasoning, on the other hand, relies on producing intermediate explicit representations such as logical forms or programs, which can then be executed against a structured knowledge base (e.g., relational database, knowledge graph, etc.) to answer questions (Gu et al., 2022; Baig et al., 2022; Zhao et al., 2023; Wang et al., 2023). These methods naturally offer better interpretability and precision thanks to the intermediate symbolic representations and their deterministic executions. However, they might be limited in expressing a broad range of natural questions in the wild depending on the semantic coverage of the underlying symbolic language and its grammar.

Neural Module Networks (Andreas et al., 2016; Gupta et al., 2019; Khot et al., 2021) have been proposed to combine neural and symbolic modality together. However, they require a symbolic language and a corresponding model that only covers limited scenarios in the specific task or domain. To apply this approach on new tasks or domains, new languages and neural modules need to be introduced. Therefore, designing a generalizable framework that uses a high-coverage symbolic expression and a flexible neural network that can be versatilely used in various scenarios becomes our goal.

In this work, we propose a **H**ybrid question **P**arser and **E**xecution framework, named HPE, for textual question answering, which combines neural and symbolic reasoning. We introduce H-

---

expression as an explicit representation of the original question, which contains primitives and operations (Liu et al., 2022). As shown in Figure 1, we consider the single-hop questions as primitives (leaves) and use symbolic operations (internal nodes) to connect them hierarchically. We introduce a semantic parser based on a seq2seq framework to parse a complex question into its corresponding H-expression in a top-down manner.

To execute an H-expression, we design a hybrid executor (H-executor), which utilizes a neural model (reader) to answer each single-hop question, and then uses deterministic rules to hierarchically compose a final answer from the single-hop answers in a bottom-up fashion. Notably, H-expression facilitates modularity in that the reader is replaceable and its training can be done globally with the massive single-hop QA data.

Our contributions can be summarized as follows:

• **Architecture**   We propose to combine the advantages of both symbolic and neural reasoning paradigms by parsing questions into hybrid intermediate expressions that can be hierarchically executed against the text to produce the final answer. Our experiments on MuSiQue (Trivedi et al., 2022b) and 2WikiMultiHopQA (Min et al., 2019) show that the proposed approach achieves state-of-the-art performance.

• **Generalizability**   End-to-end neural models are data hungry and may significantly suffer from poor generalization to unseen data, especially in limited resource scenarios. Our design, on the other hand, naturally splits the reasoning process into parsing and execution, through which it intends to disentangle learning to parse complex questions structurally from learning to resolve simple questions therein, making the process modular. Our few-shot experiments on MuSiQue and 2WikiMultiHopQA and zero-shot experiments on HotpotQA(Yang et al., 2018) and NQ (Kwiatkowski et al., 2019) suggest that even with less training data, our approach can generalize better to unseen domains.

• **Interpretability**   The execution process of our model is the same as its reasoning process. Transparency of our approach facilitates spotting and fixing erroneous cases.

## 2   Related Work

### 2.1   Neural Symbolic Systems

Gupta et al. (2019) introduce a neural module network (NMN), which solves QA using different modules by performing text span extraction and arithmetic operations. Khot et al. (2021) propose an NMN variant that decomposes a question into a sequence of simpler ones answerable by different task-specific models. Systematic question decomposition has also been explored in (Talmor and Berant, 2018; Min et al., 2019; Wolfson et al., 2020; Zhang et al., 2023).

Although our framework shares some similarities with this line of studies, there is a crucial difference in that we keep both symbolic and neural representations coincide, whereas they use a neural model to replace the non-differentiable symbolic representation to be able to train end-to-end. Furthermore, our main contribution is the use of H-expressions and the answer backtracking procedure for QA and not the question decomposition.

### 2.2   Chain-of-Thought Reasoning

A series of prior studies focus on generating explanations, which can be viewed as reasoning chains. The methods proposed in (Yavuz et al., 2022; Latcinnik and Berant, 2020; Jhamtani and Clark, 2020) formulate the multi-hop QA as single sequence generation, which contains an answer along with its reasoning path. Along this line, Large Language Models (LLMs) have recently shown its capability to answer complex questions by producing step-by-step reasoning chains, known as chains-of-thought (CoT), when prompted with a few examples (Wang et al., 2022; Zhou et al., 2022; Wei et al., 2022; Lyu et al., 2023) or even without any example (Kojima et al., 2022; Kadavath et al., 2022).

Even though the generated reasoning path by these methods may provide some explanation on how the question being solved, there is no guarantee that the answer is indeed generated by the predicted reasoning path. Furthermore, CoT reasoning paths are aimed at providing more context for the LLMs to be able to locate the right answer. This is different from our objective where our symbolic representation gives a deterministic and hierarchical reasoning (execution) path to derive an answer.

## 3   Approach

We formulate textual question answering as the task of answering a question $q$ given the textual evidence provided by a set of passages. We assume access to a dataset of tuples $\{(q_i, a_i, P_i)\}_{i=1}^n$, where $a_i$ is a text string that defines the correct answer to question $q_i$ with $P_i$ being the passage set.

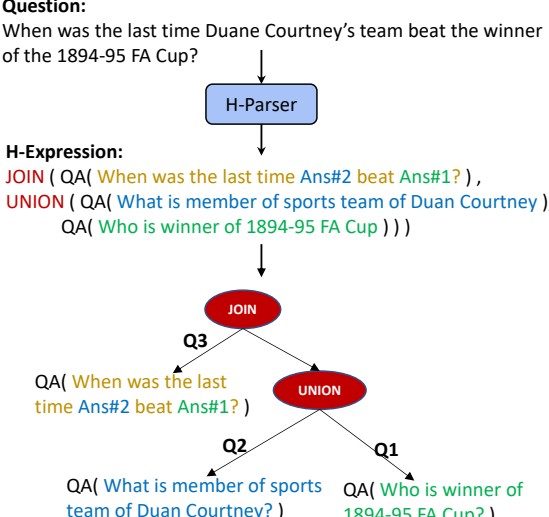

**Question:**
When was the last time Duane Courtney's team beat the winner of the 1894-95 FA Cup?

↓

H-Parser

↓

**H-Expression:**
JOIN ( QA( When was the last time Ans#2 beat Ans#1? ) ,
UNION ( QA( What is member of sports team of Duan Courtney ),
QA( Who is winner of 1894-95 FA Cup ) ) )

↓

JOIN
Q3
QA( When was the last time Ans#2 beat Ans#1? )

UNION
Q2          Q1
QA( What is member of sports team of Duan Courtney? )     QA( Who is winner of 1894-95 FA Cup? )

Figure 2: An overview of parsing with H-parser, which involves translating the input question into an H-expression, and subsequently reshaping it into a tree-structure, facilitating the determination of the node's execution sequence.

In this work, we cast the task as question parsing with hybrid execution. For a given question $q_i$, a question parser (H-parser) is tasked to generate the corresponding H-expression $l_i$ (Sec. 3.1). This along with the supporting passage set $P_i$ are then given to the execution model (H-executor) to generate an answer (Sec. 3.2). The H-executor uses a reader for simple question answering and executes the symbolic operations based on answers in a bottom-up manner to get the final prediction.

### 3.1 H-parser

To improve the compositional generalization, we follow (Liu et al., 2022) to define H-expression as a composition of primitives and operations.

**H-expression Grammar** We consider single-hop questions as primitives, which are the atomic units that form a complex question. We use operations to represent relations between primitives. We consider eight types of operations: JOIN, UNION, AND, COMP_=, COMP_>, COMP_<, SUB, and ADD. Each is a binary operation that takes two operands as input, written as OP [q2, q1], where each of q1 and q2 can be a single-hop question or result of another operation. In the execution step, q1 is executed first, then q2. The operations can be hierarchically combined into more complex one, for example, JOIN [q3, UNION [q2, q1]]. For a single-hop question, its H-expression is the question itself.

We describe the operations in Table 1 with an

example. Specifically, the JOIN[q2(Ans#1), q1] operation is used for a *linear-chain* reasoning type — q1 is an independent question that can be answered directly, while q2's execution depends on q1's answer. In the execution step, the operations will be executed in a sequence: first q1, then the answer of q1 will be used to replace the placeholder (Ans#1) in q2. The AND [q2, q1] and UNION [q2, q1] operations respectively return the common answers or all of the answers for q2 and q1; see Figure 2 for an example.

The COMP_= [q2, q1] operation is used to determine if the answers of q2 and q1 are equal and it returns "Yes" or "No". The COMP_< [q2,q1] and COMP_> [q2, q1] operations respectively select the question entity corresponding to the smaller or the bigger answer of q2 and q1. Finally, the SUB and ADD are numeric operations and perform subtraction and addition, respectively.

**H-expression Generation** The semantic parsing process of queries over knowledge bases or databases typically needs to consider the background context to match natural questions to their corresponding logical forms with the specified schema, which is a necessary condition to execute in knowledge base (Ye et al., 2022) or table (Lin et al., 2020). However, in textual QA, the question parsing process should be context-independent as we want the meaning of the original question and the H-expression to be equivalent without any additional information from the context.

Our parser is a Seq2Seq model that takes a natural question $q$ as input and generates its H-expression $l$ as output. Seq2seq formulations have been successfully used for parsing tasks (Vinyals et al., 2015). We use a T5 model (Raffel et al., 2020) as the basis of our parser, as it demonstrates strong performance on various text generation tasks. We train the model by teacher forcing – the target H-expression is generated token by token, and the model is optimized using cross-entropy loss. At inference time, we use beam search to decode the top-$k$ target H-expression in an auto-regressive manner. It is easy to transform the nested H-expression to a binary tree structure in a top-to-down manner, where the primitives constitute the leaf nodes and the internal nodes represent the deterministic symbolic operations.

| Operation & Return type | Description & Example |
|---|---|
| JOIN[ q2(Ans#1), q1] Text span | q2's execution depends on q1's answer to replace the placeholder "Ans#1". Finally returns q2's answer. |
| | Question: Where was the birth place of film The Iron Man director? |
| | H-expression: JOIN[ Where is Ans#1's place of birth?, Who is director of The Iron Man? ] |
| | Return: New York |
| UNION[ q2, q1] Dictionary | Executes q2 and q1 simultaneously and returns a dictionary as {Ans#1: a1, Ans#2: a2}, where a1/2 is the answer of q1/2. |
| | Question: Which state is Horndean located in and what is McDonaldization named after? |
| | H-expression: UNION[ Which state is Horndean located in?, What is McDonaldization named after? ] |
| | Return: {Ans#1:McDonald's , Ans#2:England } |
| AND[ q2, q1] Text spans | Executes q2 and q1 simultaneously and returns intersection of q2 and q1's answers. |
| | Question: Which former member of the Pittsburgh Pirates was nicknamed "The Cobra"? |
| | H-expression: AND[ Who is the former member of the Pittsburgh Pirates?, Who was nicknamed "The Cobra"? ] |
| | Return: Dave Parker |
| COMP_=[ q2, q1 ] Yes/No | Compare if the answers of q2 and q1 are equal. |
| | Question: Are North Marion High School (Oregon) and Seoul High School both located in the same country? |
| | H-expression: COMP_=[ Which is country of North Marion High School (Oregon)?, Which is country of Seoul High School? ] |
| | Return: No |
| COMP_<[q2, q1] COMP_>[q2, q1] Main Entity in q2/q1 | Compare the answers of q2 and q1 and return the main entity of q2 or q1. |
| | Question: Which film was came out first, Blind Shaft or The Mask of Fu Manchu? |
| | H-expression: COMP_<[ When is publication date of Blind Shaft?, When is publication date of The Mask of Fu Manchu? ] |
| | Return: The Mask of Fu Manchu |
| SUB[ q2, q1 ] Number | Subtract q2's numeric answer from q1's answer. |
| | Question: How many years does Giuseppe Cesari live? |
| | H-expression: SUB[ When does Giuseppe Cesari dead?, When does Giuseppe Cesari born? ] |
| | Return: 72 |
| ADD [ q2, q1 ] Number | Add q2's and q1's numeric answers. |
| | Question: How many siblings does Mary Shelley have? |
| | H-expression: ADD[ How many sisters does Mary Shelley have?, How many brothers does Mary Shelley have? ] |
| | Return: 4 |

Table 1: Operations defined in our H-expressions and its corresponding example of question, H-expression and return; q2 and q1 are single-hop natural questions.

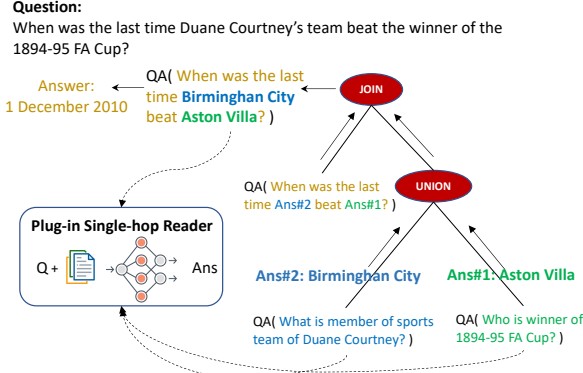

Figure 3: An overview of the H-executor. It involves execution of the question node and utilization of a plug-in neural reader to provide answer feedback for each question. Subsequently, the deterministic symbolic interpreter executes the expression to yield the final answer.

## 3.2 H-executor

Unlike the execution in databases or knowledge bases which is fully program-based, our execution has both a neural component and a symbolic component. All the primitives in the H-expression tree are executed by a neural reader and the results are aggregated by the symbolic rules expressed by the operations in the internal nodes.

Specifically, the H-executor traverses from the rightmost primitive, followed by its parent node and the left branch recursively. This is similar to in-order traversal (with opposite leaf order). As shown in Figure 3, the question "*who is winner of 1894-95 FA Cup*" is the first primitive to be executed and the single-hop reader is called to answer it, which yields the answer "*Aston Villa*". Then internal node operation UNION is visited, which stores "*Aston Villa*" as *Ans#1*. Then the left primitive "*what is member of sports team of Duane Courtney*" is visited, and the single-hop reader is called again to answer this. The reader yields "*Birminghan City*", which is stored as *Ans#2*. Operation JOIN is visited next, which replaces the placeholder *Ans#1* and *Ans#2* with the stored answer and produce a new primitive "*When was the last time Birminghan City beat Aston Villa*". This resulting primitive is answered by the single-hop reader, which predicts the final answer "*1 December 2010*".

**Plug-in Reader Network** The reader network serves as a plug-in fasion that can be replaced with any model. In our experiment, we commence with generative encoder-decoder model, FiD (Izacard and Grave, 2021) and further extend it by experimenting with various versions of FiD, each possessing distinct input and output configurations. Each supporting passage is concatenated with the question, and processed independently from other

passages by the encoder, which makes it efficient. The decoder attends to the concatenation of all encoded representations from the encoder. To distinguish different components, we add the special tokens `question:`, `title:` and `context:` before the question, title and text of each passage. Note that the reader network is detachable and may be replaced by any generative or extractive reader model at our choice.

Our assumption is that the single-hop questions are easier to answer and it is feasible to have a global single-hop reader, which can be adapted to an unseen data. To reduce training cost, we first train a T5-large in the reading comprehension setting, with one positive passage. For this, we leverage the large-scale QA pairs from Probably-Asked Questions/PAQ (Lewis et al., 2021). Then we use the trained T5-large model to initialize the FiD model and further train the model using training sets of TriviaQA (Joshi et al., 2017), SQuAD (Rajpurkar et al., 2016), BoolQ (Clark et al., 2019) in a multiple passage setting (with one positive passage and nineteen negative passages). We believe, our model, being trained on multiple datasets, can be used to unseen questions in a zero-shot manner. It can also boost the performance when used to initialize in a fine-tuning setting.

## 4 Experiments

We conduct experiments on two multi-hop multi-passage textual QA datasets, MuSiQue and 2Wikimulti-hopQA, which contain complex questions and the corresponding decomposed simple questions for the supervised setting. We also test models' generalization on the few shot setting using 5-20% of the training data. In real scenarios, neither the decomposed questions nor the complexity of questions is known. Therefore, we also investigate our models under the zero-shot setting on both complex (HotpotQA) and simple (NQ) QA datasets. In the end, we carry out a case study to show the interpretability of our framework.

### 4.1 Supervised Experiments

#### 4.1.1 Datasets

We describe each dataset and then explain how to convert the original data into the training format for both question parsing and execution.

**MuSiQue** (Trivedi et al., 2022b) contains multi-hop reasoning questions with different number of hops and question entities which can be asked from 20 supporting passages. It contains 19,938, 2,417, and 2,459 instances for train, dev and test sets, respectively, with *2hop1* (meaning questions with 2 hops and 1 entity), *3hop1*, *4hop1*, *3hop2*, *4hop2* and *4hop3* reasoning types.

**2Wikimulti-hopQA** (2WikiQA) (Ho et al., 2020) requires models to read and perform multi-hop reasoning over 10 passages. Three types of reasoning are included, namely *comparison*, *bridge*, and *bridge-comparison*. It contains 167,454, 12,576, and 12,576 for train, dev and test, respectively.

For each complex question, MuSiQue provides a reasoning type and the decomposed single questions with their answers. We use JOIN operation to combine linear-chain type questions together and use AND to combine intersection type questions. In 2WikiQA, we use evidences (in form of triplet <*subject*, *relation*, *object*>) and reasoning type to create the H-expression. We first convert the subject and relation into natural questions using templates and the object is the answer of this natural question. Then, we use the operation to combine those single-hop questions into an H-expression based on their reasoning type. In Table 1, we show a few examples of complex questions and their corresponding H-expressions; see Appendix A for more examples.

**Evaluation Metrics.** We use official evaluation scripts for each dataset with two metrics to measure answer exact match accuracy (EM) and answer token-level accuracy (F1).

#### 4.1.2 Baselines

Press et al. (2022) and Trivedi et al. (2022a) make use of large language models like GPT-3 (Brown et al., 2020). They iteratively generate an answerable question, use retrieval to get supporting passages, and answer the question based on the retrieved passages. SA (Trivedi et al., 2022b) is the state-of-the-art model on the MuSiQue dataset, which first uses a RoBERTa based (Liu et al., 2019) ranking model to rank supporting passages and then uses an End2End reader model to answer complex questions using the top-k ranked passages. EX(SA) (Trivedi et al., 2022b) decomposes a complex question into single-hop questions and builds a directed acyclic graph (DAG) for each single-hop reader (SA) to memorize the answer flow. NA-Reviewer (Fu et al., 2022) proposes a reviewer model that can fix the error prediction from incorrect evidence. We include FiD (Izacard and Grave,

| | | MuSiQue | | 2WikiQA | |
|---|---|---|---|---|---|
| | | **EM** | **F1** | **EM** | **F1** |
| Large LM | Self-ask + Search (Press et al., 2022) | 15.2 | - | 40.1 | - |
| | IRCoT (Trivedi et al., 2022a) | - | 35.5 | - | 65.2 |
| SOTA | SA (Trivedi et al., 2022b) | - | 52.3 | - | 79.5 |
| | EX(SA) (Trivedi et al., 2022b) | - | 49.0 | - | 71.2 |
| | NA-Reviewer (Fu et al., 2022) | - | - | 76.9 | 82.3 |
| End2End | FiD | 37.6 | 45.3 | 76.9 | 80.8 |
| | $FiD_{+PT}$ | 40.0 | 48.8 | 78.8 | 83.0 |
| | $FiD_{LF->Ans}$ | 36.1 | 44.8 | 76.9 | 80.5 |
| | $FiD_{CQ->LF+Ans}$ | 33.7 | 42.1 | 74.2 | 77.6 |
| Ours | HPE | 42.9 | 50.1 | 80.1 | 84.5 |
| | $HPE_{+PT}$ | **45.5** | **53.7** | **84.7** | **87.7** |

Table 2: Answer Exact match (EM) and F1 scores on dev/test split of MuSiQue and 2WikiQA. PT represents pre-training on reader network. The methods in Large LM and SOTA are reported from the previous work. The methods in End2End is implemented by us following the training details in the paper.

2021) as the baseline End2End reader model. In the original FiD, it takes the question as well as the supporting passages as input, and generates the answer as a sequence of tokens. Moreover, we propose two variants of FiD to compare the influence using H-expression: one uses H-expressions as the input, instead of original questions, to generate answers (referred to as $FiD_{LF->Ans}$), and the other uses questions as input to generate both H-expressions and answers (referred to as $FiD_{CQ->LF+Ans}$).

### 4.1.3 Implementation Details

We describe fine-tuning details for question parsing and single-hop reader models in Appendix B.

**Pre-training (PT)** To pretrain the single-hop reader, we use a subset of PAQ (Lewis et al., 2021) consisting of 20M pairs, which is generated based on named entities and the greedy decoded top-1 sequence with the beam size of 4. We train a T5-large model for 400k steps, with one gold passage, maximum length of 256 and batch size of 64. Then we initialize FiD with the PAQ pre-trained model and further train it for 40k steps, with batch size of 8 and 20 supporting passages, on the combined training sets of TriviaQA (Joshi et al., 2017), SQuAD (Rajpurkar et al., 2016) and BoolQ (Clark et al., 2019). Our code is based on Huggingface Transformers (Wolf et al., 2019). All the experiments are conducted on a cloud instance with eight NVIDIA A100 GPUs (40GB).

### 4.1.4 Fine-tuning Results

We present our main results on MuSiQue and 2WikiQA in Table 2. We observe that Self-ask and IRCoT, which are based on large language models and search engines, underperform most supervised models. This indicates that multi-hop multi-paragraph question answering is a difficult task, and there still has an evident gap between supervised small models and large models with few-shot or zero-shot.

Moreover, our framework outperforms the previous SOTA methods on both datasets. We notice that the baseline EX(SA) underperforms SA by a large margin, but our HPE outperforms FiD by 5.3% on MuSiQue EM. This shows the difficulty to build a good H-expression and executor. Moreover, EX(SA) gets a bad performance on 2WikiQA, which shows that using DAG to represent the logical relationship between sub-questions is not adaptable to any reasoning type. Compared with the End2End baseline (FiD) that our model is built on, our framework with an explicit representation performs much better.

As for $FiD_{LF->Ans}$ and $FiD_{CQ->LF+Ans}$, using H-expression as the input or output of the Seq2Seq model, expecting this facilitates the model to capture the decomposition and reasoning path in an implicit way, does not help the model. This suggests that only the proposed execution method can help the model capture the logical reasoning represented in the H-expression.

### 4.2 Few-shot Results

To illustrate the generalization ability of our framework, we show the analysis of our method under the few-shot setting in Table 3. We run three experiments, random sampling 5, 10, and 20 percent-

| | Method | MuSiQue | | 2WikiQA | |
|---|---|---|---|---|---|
| | | EM | F1 | EM | F1 |
| 5% | FiD | 29.7 | 38.7 | 55.6 | 60.6 |
| | HPE$_{FiD}$ | **34.6** | **42.3** | **67.0** | **72.3** |
| 10% | FiD | 31.2 | 40.5 | 58.3 | 63.0 |
| | HPE$_{FiD}$ | **35.2** | **43.3** | **68.3** | **73.8** |
| 20% | FiD | 31.9 | 41.0 | 69.1 | 73.5 |
| | HPE$_{FiD}$ | **36.4** | **44.2** | **73.0** | **78.0** |

Table 3: Few-shot setting Exact match (EM) and F1 scores on test/dev split of the MuSiQue and 2WikiQA.

| | HotpotQA | |
|---|---|---|
| | EM | F1 |
| Standard (Yao et al., 2022) | 28.7 | - |
| CoT (Wei et al., 2022) | 29.4 | - |
| FiD (PT) | 32.5 | **44.7** |
| HPE(PT) | **32.6** | 43.4 |
| Union HPE+FiD | 47.2 | 55.1 |
| Supervised SoTA | 72.3 | 84.9 |

Table 4: Zero-shot performance on HotpotQA. Standard and CoT are prompted method using large language model like GPT3 (Brown et al., 2020).

age of the training data. We use the End2End FiD model as the baseline, which inputs complex questions and generates the answers as token sequences. In 5% of MuSiQue dataset, it shows that our framework obtains a 4.9% absolute gain on MuSiQue EM score in comparison to the FiD model. Moreover, with 20% MuSiQue training data, our framework achieves 36.4 EM, which is a comparable performance with FiD trained on full-data (37.6 EM). Similar trends are also observed on 2WikiQA. In summary, the overall experiment shows that our model has better generalization ability than the End2End model, which is obtained by decomposing complex questions into single-hop ones and representing in H-expressions.

## 4.3 Zero-shot Results

We expect the H-parser to work well on questions of varying levels of complexity. To verify this, we test the models on two benchmarks HotpotQA and Natural Questions without any tuning. The former does not contain any decomposed questions, and the latter contains common real-world questions.

### 4.3.1 Dataset

**HotpotQA** we use the distractor setting (Yang et al., 2018) that a model needs to answer each question given 10 passages. To produce correct answer for a question, the dataset requires the model to reason across two passages. Note that two main reasoning types *bridge* and *comparison* in HotpotQA are included in MuSiQue and 2WikiQA.

**Natural Questions (NQ)** (Kwiatkowski et al., 2019) contains open-domain questions collected from Google search queries. Usually, NQ is treated as a simple question dataset and previous works usually use End2End multi-passage reader like FiD. However, we argue that certain questions in NQ involve multi-hop reasoning and the model performance can be improved by decomposing them into

single-hop questions.

### 4.3.2 Global Question Parser

To seamlessly generate H-expressions on unseen questions, we need a global question parser. This question parser can understand the complexity of the question, which means it can decompose a complex question into several simple questions and keep the simple question as is. To get a global question parser, we train a pretrained generative model T5 (Raffel et al., 2020) to convert questions to H-expressions using MuSiQue and 2Wikimulti-hopQA datasets. As the two datasets are not the same size, we categorize the complex question based on their reasoning type and sample the same amount of data for each category. To endow the model with the ability of understanding question complexity, we also use the simple questions in those datasets (the H-expression of a simple question is itself). Moreover, we decouple the composition of complex H-expressions into a few of simple H-expressions to ensure the coverage of all levels of complexity.

### 4.3.3 Zero-Shot Results on HotpotQA

We show the HotpotQA results in Table 4. We use FiD pre-trained on PAQ and TriviaQA, SQuAD and BoolQ as our zero-shot reader. Our framework outperforms both Standard and CoT, using prompt-based large language models. This shows that with the hybrid question parsing and execution framework, a small language model is generalizable on unseen questions. Compared with FiD (PT), we get a comparable performance. But checking the union of HPE and FiD, which takes the correct predictions from both methods, we find 15% absolute gain can be obtained. This shows that HPE correctly answers around 15% of questions that FiD predicts incorrectly, with the help of the question decomposition and symbolic operation. On the other hand, we conjecture that the reason that HPE

wrongly predicts some questions is that the global question parser fails to generate H-expression correctly. Hence, it is worth exploring how to design a generalizable global question parser in future work.

### 4.3.4 Results on NQ

We use the global question parser to decompose NQ question in a zero-shot manner. If a question is recognized as single-hop reasoning and cannot be further decomposed, the parser will keep the question unchanged. We use the DPR model (Karpukhin et al., 2020) to retrieve the Top-20 documents from Wikipedia as the supporting documents. Among the 8k dev set examples, 32 questions have been decomposed into single-hop questions with the logical operations and the rest are left as is. For example, a question "when did the last survivor of the titanic die" is converted into the H-expression "JOIN [when did A1 die, who was the last person to survive the titanic]". The result in Table 5 shows that HPE can handle questions of different complexity levels and will not degenerate on simple questions.

|          | EM       | F1       |
|----------|----------|----------|
| FiD (FT) | 51.4     | 56.2     |
| HPE (FT) | **51.7** | **56.3** |

Table 5: Answer exact match (EM) and F1 scores on dev split of the simple QA NQ.

### 4.4 Ablation Study

**Impact of H-parser**   We show the performance of different H-parsers. Table 6 shows using T5-large rather than T5-base, we can get around 2 to 4 percent performance improvement on both datasets. Compared to the result using gold H-expression, there is more room for improvement on the MuSiQue dataset. This might also be the case as the questions in MuSiQue are generally more complex than 2WikiQA.

|          | MuSiQue | | 2WikiQA | |
|----------|------|------|------|------|
|          | EM   | F1   | EM   | F1   |
| Gold     | 50.2 | 57.5 | 83.6 | 86.5 |
| T5-base  | 39.8 | 46.5 | 78.9 | 82.3 |
| T5-large | **42.9** | **50.1** | **80.1** | **84.5** |

Table 6: Different question parsers and the gold H-expression impact on Answer EM and F1 on MuSiQue and 2WikiQA under same FiD as the single-hop reader.

**Executability of H-parser**   We employ executability as a metric to assess the quality of the generated outputs. In Table 7, we showcase the rate at which the T5-large model generates executable logical forms within the Top-k decoded list. As we can see, the Top-1 logical forms demonstrate a high executable rate ($96\%$ - $98\%$), underscoring the model's expertise in producing syntactically correct generations. As the beam size broadens, majority of the Top-k logical forms prove to be executable, greatly enhancing our inference process.

|        | MuSiQue | 2WikiQA |
|--------|---------|---------|
| Top-1  | 95.6    | 98.3    |
| Top-10 | 99.7    | 100.0   |

Table 7: The execution rate of the H-parser.

**Impact of H-executor**   Our hybrid executor is combined with symbolic operations and replaceable reader network. We analyze the influence of different reader networks to the final performance and experiment with different versions of FiD. SupportFiD generates both answers and the supporting document titles. SelectFiD is a two-step method that first uses a RoBERTa-based (Liu et al., 2019) ranking model to predict the Top-5 relevant documents and feeds them into FiD to generate the answer. From results in Table 8, we can see that a better single-hop reader produces better performance on MuSiQue. The improvement on single-hop reader translates to a significant performance boost on complex questions.

### 4.5 Case Study

In this section, we analyze the error cases. Moreover, we show the performance under each reasoning type on MuSiQue and 2WikiQA in Appendix C. In the end, we show a case of how our model reasons on a complex question in Appendix D.

**Error Analysis**   There are two types of errors of our model prediction. One is the error from the semantic parsing of the H-expression. The other is the error from the single-hop question answer on H-execution stage. The percentage of the first type of error is $67\%$ and the second type is $33\%$ on the MuSiQue dataset.

Even though our model boasts a high execution rate, as highlighted in the table, it is not without challenges. In particular, the H-expression generated by our model might not correspond with the

| | | FiD(T5-base) Ans EM/F1 | FiD(T5-large) Ans EM/F1 | FiD(PT) Ans EM/F1 | SupportFiD(PT) Ans EM/F1 | SP EM/F1 | SelectFiD(PT) Ans EM/F1 | SP EM/F1 |
|---|---|---|---|---|---|---|---|---|
| MuSiQue | SQ | 64.9/70.7 | 68.5/74.9 | 73.3/79.5 | 72.3/78.5 | **78.6/92.2** | **76.8/82.6** | 73.8/90.2 |
| | CQ | 34.9/44.6 | 42.9/50.1 | 45.5/53.7 | 43.8/53.4 | **41.7/72.1** | **45.9/54.8** | 39.2/70.5 |

Table 8: EM and F1 scores of Answer and Support Passage on MuSiQue using different reader models. SQ represents simple question and CQ represents complex question.

reasoning type of the gold standard, thus failing to yield the correct final answer. To illustrate, if the gold H-expression is a 3-Hop while our model produces a 2-Hop, it indicates a missed decomposition into two distinct questions. As a result, the necessary context to answer the question is absent. Addressing this issue in an open-domain setting (Liu et al., 2021) could enhance the likelihood of accurately responding to more intricate questions.

As the number of hops increases, our model might face exposure bias (Bengio et al., 2015), where an error in one step can affect the final answer. However, once we identify the error's source, we can correct it, whether it's an incorrect H-expression or a single-hop answer at H-execution stage. This exposure bias can also be mitigated using a beam search (Wiseman and Rush, 2016), generating multiple answers at each step and selecting the highest-scoring one.

## 5 Conclusion

We propose HPE for answering complex questions over text, which combines the strengths of neural network approaches and symbolic approaches. We parse the question into H-expressions followed by the hybrid execution to get the final answer. Our extensive empirical results demonstrate that HPE has a strong performance on various datasets under supervised, few-shot, and zero-shot settings. Moreover, our model has a strong interpretability exposing its underlying reasoning process, which facilitates understanding and possibly fixing its errors. By replacing our text reader with KB or Table based neural network, our framework can be extended to solve KB and Table QA.

## 6 Acknowledge

The authors extend their appreciation to the members of the Salesforce AI Research team for their constructive discussions. We are also grateful to the anonymous reviewers for their invaluable feedback. Special thanks are reserved for late Prof. Dragomir Radev, whose invaluable guidance and profound insights have been instrumental throughout the course of this research.

## Limitations

We acknowledge that our work could have the following limitations:

• Even if the defined H-expression can be used on various reasoning types and different text question answering datasets, it is not mature to be used to any type of reasoning. When the new reasoning type comes, we need to retrain the question parser. To solve the new reasoning type question, we plan to take advantage of in-context learning in a large language model to generate H-expression as future work. It's worth mentioning that our executor can be easily adapted to new reasoning types by adding new symbolic rules and the reader network doesn't need to be retrained.

• As mentioned in the error analysis section, the bottom-up question answering process could suffer from exposure bias since the next step question answering may depend on the previous predicted answers. To deal with this limitation, we anticipate that generating multiple answers using beam search in each step may greatly solve this issue. Since predicted candidates by current reader models have a strong lexical overlap, general beam search needs to be revised to provide a sufficient coverage of semantic meanings. We leave it for future work.

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

## A H-expression Examples of Musique and 2WikiQA

In table 9 and 10, we show the H-expression examples and parsing tree under each reasoning type of Musique and 2WikiQA.

## B Supervised Training Details

To train the question parser, we initiate H-parser using T5-large model. We trained it with batch size of 32 with a learning rate of 3e-5 for 20 epochs on both MuSiQue and 2WikiQA. We selected the model weight based on evaluating the H-expression exact match. We base our reader network FiD on T5-large. We use 20 passages with maximum length of 256 tokens for input blocks on MuSiQue dataset and use 10 passages with 356 tokens as text length on the 2WikiQA dataset. We trained the reader model with a batch size of 8 with a learning rate of 5e-4 for 40k steps.

## C Performance of each Different Reasoning Type

We represent the Answer F1 performance under different reasoning types on both MuSiQue and 2WikiQA in Figure 5. Our hybrid question parsing and execution model performs significantly better than directly getting the answer model in both QA showing that the advantage of delegating semantic parsing to solve complex textual questions. In MuSiQue, for the relevant simple reasoning types (2hop, 3hop1), our model outperform FiD by a great margin. For complex reasoning types (3hop2, 4hop1, 4hop2 and 4hop3), our model gets lower performance compared with the simple reasoning types because the exposure bias issue becomes worse with the step of reasoning increase. But it still has a equivalent or better perform comparing End-to-End FiD. In 2WikiQA, our model performs best on all four reasoning type. Especially on the most complex type bridge comparison, our framework greatly outperform, which shows using deterministic symbolic representation is more robust to produce a correct answer.

## D A case study of how HPE reasoning

In Figure 4, we show an example that FiD predicts a wrong answer but our model correctly predicts. Given a complex question, our framework first parses the complex question into H-expression. Then hybrid executor will convert the binary tree

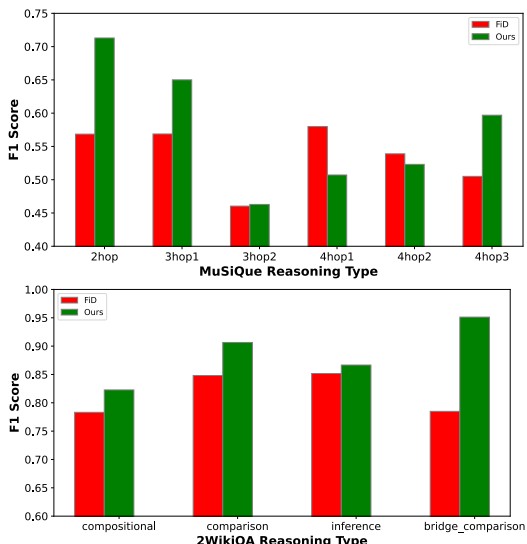

Figure 4: 3hop2-type MuSiQue question example and how our framework finds the final answer.

from the H-expression, where operation and natural sub-question as its nodes. H-executor parses the binary tree from the rightmost left node to the left and upper layer with considering the operation.

At each question node, the reader neural network will take sub-questions and multiple paragraphs as input to generate the sub-answers. We store the sub-answer in memory for later substitution of the placeholder. For example, Q3 will be rewritten in A1 with the answer of Q1 (the Republicans) and A2 with the answer of Q2 (Senate) as the new question Q3' "when did Senate take control of the Republicans". The final answer is obtained by answering Q3'.

Figure 5: Answer F1 score on each reasoning type on MuSiQue and 2WikiQA.

**Type:** 2-Hop

**Question:** Who is the deputy prime minister of the country that encompasses **Inagua National Park**?

**H-expression:** JOIN [ Who is the deputy prime minister of the Ans#1 , What is country of **Inagua National Park** ]

**Tree:**

```
                                    JOIN
                         ┌───────────────────────┐
       Who is the deputy prime minister of the Ans#1    What is country of Inagua National Park
```

---

**Type:** 3-Hop 1-Entity

**Question:** When did the greek orthodox church split from the religious institution located in the city where the creator of **The Last Judgment** died?

**H-expression:** JOIN [ When did the greek orthodox church split from Ans#2? , JOIN [ In what city did Ans#1 die? , Who is creator of **The Last Judgment** ] ]

**Tree:**

```
                                    JOIN
                         ┌───────────────────────┐
       When did the greek orthodox church split from Ans#2    JOIN
                                                    ┌──────────────────┐
                                   In what city did Ans#1 die?    Who is creator of The Last Judgment
```

---

**Type:** 3-Hop 2-Entity

**Question:** When did the capitol of Virginia move from **Robert Banks**' birthplace to the town **WTVR-FM** is licensed in?

**H-expression:** JOIN [ When did the capital of virginia moved from Ans#2 to Ans#1 , UNION [ What town is **WTVR-FM** liscensed in? , What is place of birth of **Robert Banks** ] ]

**Tree:**

```
                                    JOIN
                         ┌──────────────────────────┐
       When did the capital of virginia moved from Ans#2 to Ans#1    UNION
                                                          ┌──────────────────────┐
                                        What town is WTVR-FM liscensed in?    What is place of birth of Robert Banks
```

---

**Type:** 4-Hop 1-Entity

**Question:** When did the civil war start in the country whose capitol was home to the man after whom **Korolyov** was named?

**H-expression:** JOIN [ When did the civil war in Ans#3 start , JOIN [ Ans#2 is the capital city of which country , JOIN [ What is residence of Ans#1 , **Korolyov** is named after What ] ] ]

**Tree:**

```
                                    JOIN
                         ┌──────────────────┐
       When did the civil war in Ans#3 start    JOIN
                                      ┌──────────────────────┐
                        Ans#2 is the capital city of which country    JOIN
                                                            ┌──────────────────┐
                                          What is residence of Ans#1    Korolyov is named after What
```

---

**Type:** 4-Hop 3-Entity

**Question:** When did **Muslim armies** invade the country containing **Al-Mastumah** and the country of the man who followed the reign of **Al-Mu'tamid**?

**H-expression:** JOIN [ When did muslim armies invade Ans#3 and Ans#2 , UNION [ What is country of **Al-Mastumah** , JOIN [ What is country of citizenship of Ans#1 , **Al-Mu'tamid** is followed by What ] ] ]

**Tree:**

```
                                    JOIN
                         ┌──────────────────┐
       When did muslim armies invade Ans#3 and Ans#2    UNION
                                            ┌──────────────────────┐
                              What is country of Al-Mastumah    JOIN
                                                      ┌──────────────────────────┐
                                    What is country of citizenship of Ans#1    Al-Mu'tamid is followed by What
```

Table 9: Examples of H-expression and parsing tree under each reasoning types in MuSiQue.

**Type:** Comparision

**Question:** Which film was released first, Who Is Kissing Me? or Bush Christmas?

**H-expression:** COMP < [ What is publication date of And Who Is Kissing Me? , What is publication date of Bush Christmas]

**Tree:**

```
                                    COMP_<
                        ________________|________________
          What is publication date of Who Is Kissing Me?    What is publication date of Bush Christmas?
```

---

**Type:** Bridge Comparison

**Question:** Which film has the director who was born later, Sleepers East or Leaving Fear Behind?

**H-expression:** COMP > [ JOIN [ When is date of birth of #3 , Who is director of Sleepers East ] , JOIN [ When is date of birth of #1 , Who is director of Leaving Fear Behind ] ]

**Tree:**

```
                                      COMP_>
                   ___________________|___________________
                 JOIN                                   JOIN
          ________|________                      ________|________
  When is date of birth of #3  Who is director of    When is date of birth of #1  Who is director of Leaving Fear Behind
                                  Sleepers East
```

---

**Type:** Inference

**Question:** Who is the sibling-in-law of Favila Of Asturias?

**H-expression:** JOIN [ Who is spouse of #1 , Who is sibling of Favila Of Asturias ]

**Tree:**

```
                            JOIN
                    _________|_________
          Who is spouse of #1    Who is sibling of Favila Of Asturias
```

---

**Type:** Compositional

**Question:** Where did the founder of University Of Piura die?

**H-expression:** JOIN [ Where is #1's place of death , The Universidad De Piura is founded by Who ]

**Tree:**

```
                            JOIN
                    _________|_________
          Where is #1's place of death    The Universidad De Piura is founded by Who
```

Table 10: Examples of H-expression and parsing tree under each reasoning types in 2WikiQA.