# OpenReview forum: "HPE: Answering Complex Questions over Text by Hybrid Question Parsing and Execution"
_EMNLP/2023/Conference — EMNLP 2023 Findings_

### Official Review · Reviewer_YSXk · 2023-08-04

**Soundness:** 4

**Excitement:**

3: Ambivalent: It has merits (e.g., it reports state-of-the-art results, the idea is nice), but there are key weaknesses (e.g., it describes incremental work), and it can significantly benefit from another round of revision. However, I won't object to accepting it if my co-reviewers champion it.

**Paper Topic And Main Contributions:**

The paper introduces a new framework, Hybrid Question Parser and Execution (HPE), designed to answer complex questions over text. This approach merges the capabilities of neural networks and symbolic methods, enhancing both interpretability and generalizability in textual question answering. HPE framework is composed of two primary components: the H-parser and the H-executor. The H-parser transforms complex questions into an intermediate representation known as an H-expression, which includes primitives (single-hop questions) and operations (symbolic relationships between primitives). The H-executor then processes the H-expression using a blend of deterministic rules and a neural reader network to answer the decomposed simple questions. The authors conduct comprehensive experiments on multiple datasets, demonstrating that the HPE framework surpasses existing methods in supervised, few-shot, and zero-shot settings. The framework also improves interpretability and reveals the underlying reasoning process.

**Questions For The Authors:**

A: The evaluation of the H-parser is based on the exact match score and F1, are there any other metrics worth to be considered?

**Reasons To Accept:**

- The authors propose a new framework that combines the strengths of neural networks and symbolic methods. This design allows for a more structured and interpretable approach to question answering.
- This paper presents extensive experiments demonstrating that the HPE framework outperforms existing approaches in various settings, including supervised, few-shot, and zero-shot scenarios.
- The authors show that their framework can be applied to multiple datasets, suggesting that it has broad applicability and can handle a wide range of complex questions.

**Reasons To Reject:**

- My main concern is that the pipeline approach, while offering interpretability, can lead to error propagation. If the H-parser makes a mistake in generating the H-expression, it could lead to an incorrect final answer even if the H-executor works perfectly. The paper does propose some solutions to this issue, but it remains a fundamental challenge of the pipeline approach.
- The H-expression currently consider eight types of operations. While these cover a wide range of question types, there could be complex questions that cannot be represented using these operations.


**Reproducibility:**

4: Could mostly reproduce the results, but there may be some variation because of sample variance or minor variations in their interpretation of the protocol or method.

**Reviewer Confidence:**

4: Quite sure. I tried to check the important points carefully. It's unlikely, though conceivable, that I missed something that should affect my ratings.

---

> ### Author Rebuttal · Authors · 2023-08-29
>
> We sincerely appreciate your insightful feedback. Please find our detailed responses to your comments below (RR refers to “Reason to Reject”):
>
> ### ***RR 1: If the H-parser makes a mistake in generating the H-expression, it could lead to an incorrect final answer even if the H-executor works perfectly.***
>
> We understand the concern regarding potential error propagation inherent to pipeline approaches. To provide some clarity, our H-parser exhibits a robust executable rate. Specifically, the top-1 execution rate stands at 95.6%, and with beam search, this rate rises to 99.7% for the top 10 H-expressions. It's pertinent to note that the primary discrepancy from the gold H-expression often involves a differing reasoning path. For example, a 3-hop reasoning path in the gold might be represented as a 2-hop in our generated H-expression. It leads to a problem that among all the candidate passages, there is a chance that none of them can be helpful in answering the question. As part of our ongoing research, we're looking into enhanced retrieval models and refined question decomposition techniques to mitigate this issue.
>
>
> ### ***RR 2: The H-expression currently consider eight types of operations. While these cover a wide range of question types, there could be complex questions that cannot be represented using these operations.***
>
> We recognize the limitation that the eight operations may not represent all possible complex questions. While our approach synergizes both neural and symbolic methods to address the question-answer task, capturing every facet of complex reasoning extends beyond this study's objectives. We’d like to share some of our proposals here.
>
> 1) In-context learning. We think LLMs, when provided with suitable prompts, can help expand our model's capability to generate new types of H-expressions. Our initial experiments suggest that LLMs can efficiently generate basic 2-hop H-expressions, but face challenges with more complicated structures like 4-hop expressions.
> 2) Prompt Tuning. To obtain a more specified LLM for question answering in a neural-symbolic way, we could add every operation of the corresponding reasoning type as additional soft prompt tokens.  In this way, we could learn the operations as the prompts and easily adapt to the new complex reasoning tasks.
>
> Both of these strategies offer a path toward making our framework more adaptable to various complex reasoning tasks.
>
> Input template to ChatGPT for linear-chain and intersection reasoning:
> ```
> Convert question -> H-expresion: question is the multi-hop complex question. And the H-expression contains operators and elements. Each element is a natural question. Operator is from [JOIN, AND] and each of them is a binary operator, which means it contains two elements. JOIN(QA(a) ; QA(b)) operator is for linear-chain reasoning that question a contains the placeholder (e.g. Ans#1) of question b's answer. Placeholder is to represent the answer from previous question. AND(QA(a) ; QA(b)) operator is for intersection reasoning that question a and b can be processed parallel.
> JOIN and AND can be compositioned, like JOIN(QA(c), JOIN(QA(b), QA(a))), AND(QA(c), JOIN(QA(b), QA(a))). Which composition to use, please follow the demo examples.
> For example:
> question: Who plays the creator of Sunflowers in Doctor Who?
> H-expression: JOIN( QA( who plays Ans#1 in doctor who ) , QA( Who is creator of Sunflowers ) )
> another 5 demos
> Given question: XXX,
> H-expression:
> ```
> Here are two real outputs by ChatGPT.
>
> ***Good example:***
>
> > ***Given question:*** At the end of what year did the tribes from the place where the Rossen culture is located invade the Roman Empire?
>
> > ***Gold H-expression:*** JOIN( QA( At the end of what year did the tribes from Ans#1 invade the Roman Empire? ), QA( Where is the Rossen culture located? ) )
>
> > ***ChatGPT Generated H-expression:*** JOIN( QA( At the end of which year did Ans#1 tribes invade the Roman Empire? ) , QA( Where is Rössen culture located on terrain feature? ) )
>
>
> ***Wrong example:***
>
> >***Given question:*** What is the longest home run in the history of the league that Jim Wilson's team is a part of?
>
> >***Gold H-expression:*** JOIN( QA( what is longest home runs in Ans#2 history? ) , JOIN( QA( What league was Cleveland Ans#1?) , QA( Jim Wilson is the member of which sports team?) ) )
>
> >***ChatGPT Generated H-expression:*** JOIN(QA(What is the longest home run in the history of the league that Ans#2's team is a part of), QA(Who is Jim Wilson))
>
> ### ***Question: The evaluation of the H-parser is based on the exact match score and F1, are there any other metrics worth to be considered?***
> Beyond the exact match score and F1 metrics, we've also considered BLEU and ROUGE for evaluating the H-parser's output. Specifically, we observed BLEU-2 at 79.20, BLEU-4 at 63.09, ROUGE-2 at 70.64, and ROUGE-L at 83.66 with the T5-large model. While these scores provide valuable insight into the H-parser's performance, we chose to emphasize the Answer EM and F1 metrics in our report, as they offer a more rigorous representation of our system's efficacy. We will add the BLEU and ROUGE results in the final version.

---

### Official Review · Reviewer_NMo5 · 2023-08-05

**Soundness:** 4

**Excitement:**

3: Ambivalent: It has merits (e.g., it reports state-of-the-art results, the idea is nice), but there are key weaknesses (e.g., it describes incremental work), and it can significantly benefit from another round of revision. However, I won't object to accepting it if my co-reviewers champion it.

**Paper Topic And Main Contributions:**

This work proposes a new approach called the Hybrid Parsing and Execution (HPE) framework for answering complex questions using text. It consists of a network to decompose the complex question into a semantic parse called the H-expression; where each primitive in the expression is a simple (single-hop) query. A second reader network then answers each simple question and the H-expression is resolved recursively bottom-up.

It seems to achieve SOTA on the MuSIQue and 2WikiQA datasets, as well as demonstrate strong performance in the few-shot and zero shot settings in HotPotQA.

**Questions For The Authors:**

1.  Does the non-PT version of HPE still use TriviaQA, SQuAD and BoolQ? What is the performance of an SA (PT) if you can train such a model?
2. Could you provide additional details about the CoT result on zero-shot HotPotQA? Were there exemplars used, use of self-consistency, instruction-tuned model, etc.

**Reasons To Accept:**

In my opinion, the primary strength of the work is its ability to provide an interpretable chain of reasoning in the zero-shot setting at a competitive level of performance with End2End approaches and potentially better than any other approach which can provide a chain of reasoning.

The paper is written well.

**Reasons To Reject:**

I believe there are two main problems with this work -

First, when measuring the performance against SOTA in MuSIQue and 2WikiQA, which is SA (Trivedi et al 2022b), there is a possible source of difference which does not relate to the proposed method: the additional datasets used for training the reader, which include PAQ, TriviaQA, SQuAD and BoolQ. I believe that even the non-PT version of HPE uses the latter 3 datasets, please refer to question (1). Thus, a fair comparison of performance would be SA (PT) vs HPE (PT); and looking at SA vs HPE, I suspect the performances would be more on-par. The fact that FID does just as well on zero-shot HotPotQA as HPE seems to indicate that the H-expressions might not be providing any significant performance improvement.

If that is the case, then the main contribution of this work is not the performance improvement but the interpretable chain of reasoning this method provides. If so, its main 'competitor' would be the Chain-of-Thought (CoT) approaches and its more recent advancements, which are not done justice in this work (See question 2). While theoretically HPE's chain of reasoning is deterministically executed while no such guarantee is present in LLM's, it is unclear how much of a practical problem this is. I strongly advocate for an empirical comparison against a best-effort CoT baseline to strengthen your work - meaning CoT with Self-Consistency, in-context exemplars from MuSIQue and 2WikiQA, and ideally with an instruction-tuned LLM.

**Reproducibility:**

4: Could mostly reproduce the results, but there may be some variation because of sample variance or minor variations in their interpretation of the protocol or method.

**Reviewer Confidence:**

4: Quite sure. I tried to check the important points carefully. It's unlikely, though conceivable, that I missed something that should affect my ratings.

---

> ### Author Rebuttal · Authors · 2023-08-29
>
> We would like to express our gratitude for your detailed review and feedback. We believe there might have been some misunderstandings which we hope to clarify in this rebuttal.
>
> ### Question 1(a): Does the non-PT version of HPE still use TriviaQA, SQuAD and BoolQ?
>
> Sorry that our symbols cause your confusion. We clarify that **these datasets were not utilized in the non-PT version of HPE**. Our multi-passage single-hop reader PT process has two steps. First, it uses single-passage single-hop PAQ to train a T5 model. This tuned T5 model then serves as an initialization for FiD, which undergoes further training on the multi-passage single-hop datasets TriviaQA, SQuAD, and BoolQ. Therefore, when we refer to the "non-PT version", it implies a version untouched by either PAQ or the other three datasets. More pre-training details on the reader network can be found in Section 4.1.3 and Appendix B.
>
> ### Question 1(b): What is the performance of an SA (PT) if you can train such a model?
>
> Thank you for pointing out this question. We integrated our pre-training strategy using the same collection of PT single-hop QA datasets for SA. More precisely:
> * We first pre-trained the SA model on PAQ and TriviaQA, SQuAD and BoolQ.
> * We fine-tuned the SA model on MuSiQue.
>
> However, intriguingly, the resulting model’s performance is 51.8, which is worse than non-PT 52.0(reproduced by us). This suggests that pre-training on single-hop QA datasets may impair the SA model's performance on MuSiQue. One plausible explanation is that single-hop QA tasks do not strengthen the ability of complex multi-hop reasoning required by MuSiQue. This observation amplifies the significance of our approach of offloading the complex reasoning to hybrid parsing and execution, while single-hop QA engine is all it needs. We will include these findings in our finalized version.
>
> ###  Question 2: Could you provide additional details about the CoT result on zero-shot HotPotQA?
>
> We report the CoT results from ReAct[1], which uses PaLM-540B as the LLM. Both Standard and CoT use few-shot in-context learning. For CoT with self-consistency, ReAct reported a score of 33.4. Notably, self-consistency serves as a general decoding upsampling approach that can also be applied to our HPE framework by generating multiple answers in each single-hop reader. Notably, HPE built on 770M T5-large obtains a score of 33.9 when coupled with self-consistency, which outperforms self-consistency CoT with PaLM-540B. We will incorporate these details in the finalized version.
>
> In conclusion, we are grateful for the depth and precision of your feedback. Such insights guide us in enhancing the clarity and completeness of our work. We hope that the clarifications provided address your concerns.
>
> [1]Yao, Shunyu, Jeffrey Zhao, Dian Yu, Nan Du, Izhak Shafran, Karthik R. Narasimhan, and Yuan Cao. "ReAct: Synergizing Reasoning and Acting in Language Models." In The Eleventh International Conference on Learning Representations. 2023.

---

### Official Review · Reviewer_CHKJ · 2023-08-09

**Soundness:** 4

**Excitement:**

3: Ambivalent: It has merits (e.g., it reports state-of-the-art results, the idea is nice), but there are key weaknesses (e.g., it describes incremental work), and it can significantly benefit from another round of revision. However, I won't object to accepting it if my co-reviewers champion it.

**Paper Topic And Main Contributions:**


This paper proposes a hybrid framework (HPE) that combines neural networks and symbolic reasoning for answering complex questions over text. The key ideas are:
(1) Parse questions into an intermediate representation called H-expression, which hierarchically combines simple questions using symbolic operations like JOIN, UNION, etc.
(2) Execute the H-expression using a hybrid executor - use a neural reader to answer simple questions and symbolic rules to deterministically aggregate the answers.




Limitations:

Limited coverage of reasoning types. New types require retraining the parser.
Exposure bias in multi-hop reasoning. Beam search could help but needs more exploration.
Overall, this is a novel and promising direction bringing together the strengths of neural and symbolic reasoning. The results demonstrate strong performance and the modular design provides interpretability. Addressing the limitations around adaptability and exposure bias could make this approach more robust and widely applicable.

**Questions For The Authors:**

How can the framework adapt to new reasoning types or datasets without retraining the parser? Could in-context learning help here?

Have the authors experimented with generating multiple candidate answers per step using beam search? How well does it address exposure bias?

The paper mentions replacing the text reader with a KB or table reader to extend this framework. Have the authors experimented with this? If so, how well does the framework adapt to KBQA or text-to-SQL semantic parsing tasks?

How does the framework compare to recent work on prompt-based reasoning for complex QA? Could prompting help improve the parser?

**Reasons To Accept:**

Pros:

Achieves new SOTA results on complex QA datasets MuSiQue and 2WikiMultiHopQA, outperforming prior neural and prompting-based methods.

Shows stronger generalization ability in few-shot and zero-shot settings compared to end-to-end neural models like FiD. This modularity helps disentangle parsing complex questions from answering simple ones.

Quantitative results show the framework generalizes better with less training data. This could make it more adaptable with limited labeled data.

Provides interpretability by exposing the explicit reasoning process through H-expressions. This facilitates understanding and fixing errors.

**Reasons To Reject:**

Cons:

The set of symbolic reasoning operations may not cover the full range of complex reasoning required for open-domain QA. Expanding the operations would require retraining the parser.

The parser relies on training data to generate valid H-expressions. It may fail on new reasoning types without retraining.

Exposure bias during multi-hop reasoning - an error in one step affects subsequent steps. Beam search could help mitigate this issue.


**Reproducibility:**

2: Would be hard pressed to reproduce the results. The contribution depends on data that are simply not available outside the author's institution or consortium; not enough details are provided.

**Reviewer Confidence:**

3: Pretty sure, but there's a chance I missed something. Although I have a good feel for this area in general, I did not carefully check the paper's details, e.g., the math, experimental design, or novelty.

---

> ### Author Rebuttal · Authors · 2023-08-29
>
> We would like to express our gratitude for your detailed review and feedback.
> ### Question 1: how can the framework adapt to new reasoning types or datasets without retraining the parser? Could in-context learning help here?
>
> Thank you for the insightful question. In our HPE framework, we use a T5-base or T5 large as the H-parser, which has limited model capability compared to LLMs. In this setting, to accommodate new reasoning types, we acknowledge that our current model demands retraining. Indeed, our neural-symbolic approach can be used with LLMs, where we believe adding new reasoning types could be much easier with no retraining or solely prompt tuning. Although covering the full aspects of complex reasoning through LLMs is beyond this paper's scope, we’d like to share some of our proposals here.
>
> 1) In-context learning. Yes, we believe in-context learning could help with H-expression generation. We could employ a prompt starting with the parsing task description, followed by the H-expression grammar and several demonstration examples. This setup enables the LLM to generate new H-expressions for complex questions without needing to retrain the model. We did some experiments exploring the potential of generating H-expression with in-context learning. Our result suggests that although LLM can proficiently generate basic H-expressions like 2-hop structures, it encounters limitations when dealing with more advanced composition reasoning types like 4-hop. The following is an example of in-context learning in LLMs.
>
> 2) Prompt tuning. To obtain a more specified LLM for question answering in a neural-symbolic way, we could add every operation of the corresponding reasoning type as additional soft prompt tokens.  In this way, we could learn the operations as the prompts, rather than retraining the whole model.
>
> Input template to ChatGPT for linear-chain and intersection reasoning:
> ```
> Convert question -> H-expresion: question is the multi-hop complex question. And the H-expression contains operators and elements. Each element is a natural question. Operator is from [JOIN, AND] and each of them is a binary operator, which means it contains two elements. JOIN(QA(a) ; QA(b)) operator is for linear-chain reasoning that question a contains the placeholder (e.g. Ans#1) of question b's answer. Placeholder is to represent the answer from previous question. AND(QA(a) ; QA(b)) operator is for intersection reasoning that question a and b can be processed parallel.
> JOIN and AND can be compositioned, like JOIN(QA(c), JOIN(QA(b), QA(a))), AND(QA(c), JOIN(QA(b), QA(a))). Which composition to use, please follow the demo examples.
> For example:
> question: Who plays the creator of Sunflowers in Doctor Who?
> H-expression: JOIN( QA( who plays Ans#1 in doctor who ) , QA( Who is creator of Sunflowers ) )
> another 5 demos
> Given question: XXX,
> H-expression:
> ```
> Here are two real outputs by ChatGPT.
>
> ***Good example:***
>
> > ***Given question:*** At the end of what year did the tribes from the place where the Rossen culture is located invade the Roman Empire?
>
> > ***Gold H-expression:*** JOIN( QA( At the end of what year did the tribes from Ans#1 invade the Roman Empire? ), QA( Where is the Rossen culture located? ) )
>
> > ***ChatGPT Generated H-expression:*** JOIN( QA( At the end of which year did Ans#1 tribes invade the Roman Empire? ) , QA( Where is Rössen culture located on terrain feature? ) )
>
>
> ***Wrong example:***
>
> >***Given question:*** What is the longest home run in the history of the league that Jim Wilson's team is a part of?
>
> >***Gold H-expression:*** JOIN( QA( what is longest home runs in Ans#2 history? ) , JOIN( QA( What league was Cleveland Ans#1?) , QA( Jim Wilson is the member of which sports team?) ) )
>
> >***ChatGPT Generated H-expression:*** JOIN(QA(What is the longest home run in the history of the league that Ans#2's team is a part of), QA(Who is Jim Wilson))
>
> ### Question 2: Have the authors experimented with generating multiple candidate answers per step using beam search? How well does it address exposure bias?
>
> Yes, we extended our exploration beyond beam search solely in the H-parser, delving into its integration within the H-executor as well. This approach involves utilizing beam search to generate a selection of candidate answers from the single-hop reader. In the context of HotpotQA, for instance, we applied beam search to generate the Top-10 answers, subsequently implementing a majority vote mechanism for each step. The results of this strategy led to an elevated performance from 32.6 to 33.9. This initial observation indicates that refining answer quality at each discrete step can effectively mitigate exposure bias.
>
> While the end-to-end model may not inherently suffer from exposure bias, its performance often degenerates when dealing with complex tasks due to the obscured nature of its reasoning process. Correcting an erroneous response to a complex question within an end-to-end model can be challenging. Conversely, our hybrid framework, intertwining neural and symbolic components, offers an explicit expression of the question through H-expression. Consequently, in instances of incorrect final answers, we are adept at pinpointing the erroneous sub-step of reasoning, facilitating precise corrections. The experiments and the discussions about exposure bias will be added in the revision.
>
>
> ### Question 3: The paper mentions replacing the text reader with a KB or table reader to extend this framework. Have the authors experimented with this? If so, how well does the framework adapt to KBQA or text-to-SQL semantic parsing tasks?
>
> We haven't directly tailored our framework for KBQA or text-to-SQL tasks. Instead, we're delving into a multi-source complex question-answering approach. In this paradigm, one hop might draw from textual sources for answers, whereas the next could tap into a KB or table. One of our primary challenges is the absence of a comprehensive benchmark to assess the model's performance in such settings. Our ongoing efforts are concentrated on establishing this benchmark and investigating the potential of integrated QA engines.
>
> ### Question 4: How does the framework compare to recent work on prompt-based reasoning for complex QA? Could prompting help improve the parser?
>
> In comparison to recent prompt-based reasoning approaches, such as CoT, our HPE framework shares a common methodology in decomposing complex questions into single-hop questions. However, a distinguishing feature of our approach is the utilization of symbolic operations during the reasoning process, ensuring that HPE offers more deterministic outcomes. This characteristic reduces the risk of LLM hallucinations, making our answers more reliable. When evaluated in a zero-shot setting on HotpotQA against CoT, our model, built on the 770M T5, outperforms with a score of 32.6, compared to CoT's 29.4, which uses the larger 540B PaLM. Additionally, as mentioned in Answer 1, in-context learning and prompt tuning could facilitate the parser to add new reasoning types without fully retraining.

---

### Official Review · Reviewer_5EBV · 2023-08-11

**Typos Grammar Style And Presentation Improvements:** 1. Several references need their conf…
**Soundness:** 3

**Excitement:**

3: Ambivalent: It has merits (e.g., it reports state-of-the-art results, the idea is nice), but there are key weaknesses (e.g., it describes incremental work), and it can significantly benefit from another round of revision. However, I won't object to accepting it if my co-reviewers champion it.

**Paper Topic And Main Contributions:**

This paper addresses the problem of complex question answering by applying an intermediate step which decomposes the complex question into simple logical expressions. Their approach consists of a H-parser which: 1) generates an H-expression and 2) decomposes the H-expression into simpler statements that can be executed in order. Afterward, they utilize a reader network to extract the answer for each subquestion.

The approach shows better results than SOTA our end to end problems, and the authors also analyze zero shot performance. However, the significance of the results it is not clear, and which component brings the improvement. The components of the model are pretrained in other datasets, and it is not clear if the other models are also pretrained in the same datasets. For example, FiD + Pretraining, has better performance than FiD + H-expressions.

**Questions For The Authors:**

A: Line 508-510. Which LLM did you use as a baseline to compare in zero-shot performance?
B:  When you get a wrong H-expression, how do you decompose it? Maybe I missed it, but is there any constraint for generating the H-expressions? Because if not, you may get a wrong H-expression which cannot be decomposed.
C: For the model FiD LF->Ans, how do you extract the H-expressions? If it is the same way that you utilize for your model, then what is the difference between FiD LF->Ans and your model?

**Reasons To Accept:**

- Decomposition of complex questions into simpler explainable H-expressions.
- Proposed method achieves better performance compared to SOTA and end-to-end methods.

**Reasons To Reject:**

- Analysis into how the model's components contributed to the performance compared to other models
- Limited explanation on how the approach deals with errors from the semantic parsing model

**Reproducibility:**

5: Could easily reproduce the results.

**Reviewer Confidence:**

4: Quite sure. I tried to check the important points carefully. It's unlikely, though conceivable, that I missed something that should affect my ratings.

---

> ### Author Rebuttal · Authors · 2023-08-29
>
> We genuinely thank you for the feedback and the issues you've raised. While we acknowledge and address the issues you've pointed out, we'd like to highlight that some of the major concerns have been covered in the paper and do not detract from our study's central contributions. We provide a detailed response to each point below (RR refers to "Reason to Reject", TGS refers to "Typos Grammar Style"):
>
> ### RR 1: How do the model’s components contribute to the performance compared to other models?
> Thank you for this question. In our paper, we present a synergistic integration of neural and symbolic approaches within a unified framework. We’d like to underscore the significance of distinct components across two key dimensions:
>
> 1) Effectiveness of Combined Approach: A sole reliance on neural methods proves to be insufficient. In Table 2, our integrated neural-symbolic model, HPE, achieves a F1 score of 50.1 on the MuSiQue dataset. Conversely, when evaluating FiD(LF->Ans), a configuration where the H-expression serves as the input question and the network generates the answer, we observe underwhelming performance, yielding only a 44.8 F1 score on MuSiQue. This discrepancy underscores the tangible benefits brought about by the H-executor component, which operates as a potent symbolic processor.
>
> 2) Robustness of H-parser: Detailed in Table 6 is an ablation study conducted on the H-parser module. When supplying gold H-expressions to the H-executor, the F1 scores reach 57.5 on MuSiQue and 86.5 on 2WikiQA. When we substitute the gold H-expressions with a T5-large H-parser, which generates H-expressions fed into the H-executor, the F1 scores decrease slightly to 50.1 on MuSiQue and 84.5 on 2WikiQA. This yields a difference of 7.4 on MuSiQue and 2.0 on 2WikiQA, indicating an acceptable margin between the outcomes derived from generated H-expressions and those gold labels.
>
> ### RR2 & Question B: When you get a wrong H-expression, how do you decompose it?
>
> That’s a good question. For H-parser, the executable rate of the greedy decoding H-expression is 95.6%. For further improvement, we use beam search in H-parser to generate a list of 10 H-expression candidates (mentioned in Line 240-242). Each H-expression is executed until we identify an executable one. In this way, the execution rate rises to 99.7% for the top 10, pointing to the robustness of our H-parser. To enhance clarity, we will include an executability analysis for Top-k H-expressions in the revised version.
>
>
> ### Question A: Line 508-510. Which LLM did you use as a baseline to compare in zero-shot performance?
>
> To illustrate our model's capability within a zero-shot context, we engaged in a comparative analysis of our methodology against the Standard and CoT prompting methodologies outlined in ReAct [1]. In these methodologies, the PaLM-540B model is utilized as the LLM. While our approach operates in a zero-shot paradigm, both the Standard and CoT strategies leverage few-shot in-context learning. Notably, our framework in a zero-shot setting, reliant solely on the 770M T5-large model, surpasses the few-shot results of Standard and COT with PaLM-540B when evaluated on the HotpotQA dataset. The success of our approach underscores the benefits of integrating neural networks with symbolic reasoning to answer complex questions.
>
> ### Question C: the difference between FiD LF->Ans and your model
>
> “FiD LF->Ans” means that the FiD model takes H-expression along with multi-passage context as inputs, and subsequently generates the answer as a sequence of tokens (Seq2Seq). This mechanism contrasts with our method, which capitalizes on hybrid parsing (via the H-parser) and execution (through the H-executor).
>
> ### TGS: need their conference citation instead of the arXiv version
>
> We appreciate your suggestion on correcting the citation format. We will update all references and use the correct format in the revised manuscript.
>
> We hope that our explanations effectively address your concerns and potentially lead to an adjustment of your initial assessment. Thank you for your consideration.
>
> [1]Yao, Shunyu, Jeffrey Zhao, Dian Yu, Nan Du, Izhak Shafran, Karthik R. Narasimhan, and Yuan Cao. "ReAct: Synergizing Reasoning and Acting in Language Models." In The Eleventh International Conference on Learning Representations. 2023.

---

### Meta-Review · Area_Chair_2xMi · 2023-09-16

**Recommendation:** 3

**Metareview:**

Problem: This paper addresses the problem of complex question answering.

Solution: It applies an intermediate step that decomposes the complex question into simple logical expressions. Their approach consists of an H-parser which:
 1) generates an H-expression, hierarchically combining simple questions using symbolic operations like JOIN, UNION, etc.
 2) Afterward, they utilize a reader network to extract the answer for each subquestion.


Here are the notable points of concern:

- More extensive analysis of the factors contributing to the model's success. The authors elaborate on this, and I suggest they incorporate all these details in their revised paper.
- How does the approach deal with errors from the parser? While the authors provide some statistics, the answer to this question is unclear.
- Given the model's reliance on supervised data, expanding the model to tackle more diverse questions would require retraining the parser.
- My addition to this list: The paper has no 2023 citations, which is odd. Please revise their draft with all the relevant work published this year.


All reviewers have assigned high soundness scores and overall buy the outcome. However, given the limited excitement about this work, I recommend the "findings" track.

---

### Decision · Program_Chairs · 2023-10-07

**Decision:**

Accept-Findings

**Comment:**

Problem: This paper addresses the problem of complex question answering.

Solution: It applies an intermediate step that decomposes the complex question into simple logical expressions. Their approach consists of an H-parser which:
 1) generates an H-expression, hierarchically combining simple questions using symbolic operations like JOIN, UNION, etc.
 2) Afterward, they utilize a reader network to extract the answer for each subquestion.


Here are the notable points of concern:

- More extensive analysis of the factors contributing to the model's success. The authors elaborate on this, and I suggest they incorporate all these details in their revised paper.
- How does the approach deal with errors from the parser? While the authors provide some statistics, the answer to this question is unclear.
- Given the model's reliance on supervised data, expanding the model to tackle more diverse questions would require retraining the parser.
- My addition to this list: The paper has no 2023 citations, which is odd. Please revise their draft with all the relevant work published this year.


All reviewers have assigned high soundness scores and overall buy the outcome. However, given the limited excitement about this work, I recommend the "findings" track.